# Evaluation of Pretreatment Albumin–Bilirubin Grade as a Better Prognostic Factor Compared to Child–Pugh Classification in Patients with Hepatocellular Carcinoma Receiving Transarterial Chemoembolization Combined with Radiotherapy

**DOI:** 10.3390/jpm13020354

**Published:** 2023-02-17

**Authors:** Jason Joon Bock Lee, Jun Su Park, Hyun Pyo Hong, Myung Sub Kim, Dong-Hoe Koo, Hyebin Lee, Heerim Nam

**Affiliations:** 1Department of Radiation Oncology, Kangbuk Samsung Hospital, Sungkyunkwan University School of Medicine, Seoul 03181, Republic of Korea; 2Department of Radiation Oncology, Chungnam National University Sejong Hospital, Chungnam National University School of Medicine, Sejong 30099, Republic of Korea; 3Department of Radiology, Kangbuk Samsung Hospital, Sungkyunkwan University School of Medicine, Seoul 03181, Republic of Korea; 4Division of Hematology/Oncology, Department of Internal Medicine, Kangbuk Samsung Hospital, Sungkyunkwan University School of Medicine, Seoul 03181, Republic of Korea

**Keywords:** hepatocellular carcinoma, transarterial chemoembolization, radiotherapy, albumin–bilirubin grade, Child–Pugh classification

## Abstract

This study assessed the use of pretreatment albumin–-bilirubin (ALBI) grade as a prognostic factor in patients with hepatocellular carcinoma (HCC) receiving combined transarterial chemoembolization (TACE) and radiotherapy (RT). Patients who underwent RT following TACE between January 2011 and December 2020 were analyzed retrospectively. The survival outcomes of patients in regard to the ALBI grade and Child–Pugh (C–P) classification were evaluated. A total of 73 patients with a median follow-up of 16.3 months were included. Thirty-three (45.2%) and forty patients (54.8%) were categorized into ALBI grades 1 and 2–3, respectively, while sixty-four (87.7%) and nine (12.3%) were C–P classes A and B, respectively (*p* = 0.003). The median progression-free survival (PFS) and overall survival (OS) for ALBI grade 1 vs. 2–3 were 8.6 months vs. 5.0 months (*p* = 0.016) and 27.0 months vs. 15.9 months (*p* = 0.006), respectively. The median PFS and OS for C–P class A vs. B were 6.3 months vs. 6.1 months (*p* = 0.265) and 24.8 months vs. 19.0 months (*p* = 0.630), respectively. A multivariate analysis showed that ALBI grades 2–3 were significantly associated with worse PFS (*p* = 0.035) and OS (*p* = 0.021). In conclusion, the ALBI grade could be a good prognosticator in HCC patients who were treated with combined TACE-RT.

## 1. Introduction

Hepatocellular carcinoma (HCC) is the most common histopathology among primary liver cancers and accounts for a large proportion of cancer-related mortalities worldwide [1,2]. Although the global incidence of HCC has shown mixed trends [3], the prognosis of HCC remains poor [4]. Another specific characteristic of HCC is the wide variety of available treatment options according to the tumor stage and underlying liver conditions, considering that most patients with HCC have comorbid chronic hepatitis or liver cirrhosis [5,6,7]. Although the use of the Child–Pugh (C–P) classification for liver function assessment and prognosis estimation in patients with HCC is very common [8], it has some limitations, such as the subjectivity of ascites and the encephalopathy assessment [9]. To compensate for the limitations of the C–P classification, an albumin–bilirubin (ALBI) grade as a surrogate has been under validation recently [10,11,12]. However, no universal treatment consensus has been reached to date with various staging systems and treatment guidelines from multiple academic societies [13,14].

Transarterial chemoembolization (TACE) is usually reserved for intermediate-stage HCC according to Barcelona Clinic Liver Cancer (BCLC) staging in most treatment guidelines, while systemic therapy is usually the treatment of choice for advanced disease with portal vein invasion or extrahepatic spread [13]. In addition, radiotherapy (RT) has become a possible treatment option in various clinical settings, from an early stage to metastatic HCC, with acceptable outcomes in line with the development of modern RT techniques [15]. Some studies have also suggested that TACE could be applied in combination with RT in patients with advanced HCC with good clinical outcomes [16,17]. At our institution, RT has been frequently advised after incomplete TACE in an effort to improve intrahepatic and/or extrahepatic tumor control, such as major vascular invasion, large tumor size, and regional lymph node metastases. However, among the many possible prognostic factors, evaluation of the abovementioned ALBI grade as a prognosticator for HCC patients treated with combined TACE and RT is scarce. Therefore, in this study, we aimed to investigate the role of the ALBI grade in such patients and compared it with the C–P score.

## 2. Materials and Methods

### 2.1. Patients

All patients with HCC who received TACE combined with RT between January 2011 and December 2020 at Kangbuk Samsung Hospital were reviewed. The requirement for RT as an additional treatment to TACE was determined under multidisciplinary discussion either before the administration of TACE, during the TACE procedure, or at the immediate computerized tomography or magnetic resonance imaging follow-up. The exclusion criteria were biopsy-proven histology other than HCC, the existence of distant metastasis at the time of treatment, double primary cancer either synchronously developed or requiring active treatment, prior history of abdominal RT, and an interval between TACE and RT of >24 weeks. 

### 2.2. Treatments

The detailed procedure for TACE at our institution has been described previously [18,19]. RT was performed for either curative or palliative aims according to the guidelines of the Korean Liver Cancer Association [20,21] and applied after the general condition recovery of patients with a stabilized liver enzyme found in blood chemistry evaluation. All patients were educated on respiratory control. Respiratory gating or the breath-hold technique was utilized according to the radiation oncologist’s discretion. The gross tumor volume (GTV) included viable tumor areas, tumor areas filled by lipiodol, and tumor areas showing tissue necrosis after TACE. The internal target volume (ITV) was delineated based on the tumor movement during individual respiratory phases, and the planning target volume (PTV) was defined as the ITV with a 5-mm margin. RT doses were prescribed according to the physicians’ discretion to maximize the dose delivered to the tumor while satisfying the dose constraints for normal organs, such as the remnant liver and gastrointestinal tract. 

### 2.3. Pretreatment Liver Function Assessment and Toxicity Follow-Up

An ALBI score was calculated using the following formula: (log10 bilirubin [µmol/L] × 0.66) + (albumin [g/L] × −0.0852). The ALBI grade was defined as grade 1 (ALBI score ≤ −2.60), grade 2 (ALBI score > −2.60 and ≤ −1.39), or grade 3 (ALBI score > −1.39) [11]. The C–P score and classification have been conventionally defined [22]. ALBI and C–P scores were measured within 24 h before TACE. Routine laboratory examinations other than those required by the physician were performed; however, those performed within a week before the initiation of RT and 1 or 2 months after RT were incumbent in all included cases. The incidence of radiation-induced liver disease (RILD) was evaluated during the post-treatment follow-up period up to at least 4 months after RT, whenever possible. RILD was defined as either classic RILD, an increase in the alkaline phosphatase (ALP) level more than two-fold of the upper normal limit, or non-classic RILD, the elevation of the transaminase level above fivefold of the upper normal limit [23,24]. Patients with liver function deterioration with evidence of disease progression were excluded.

### 2.4. Statistical Analysis

Patient characteristics according to the ALBI grade were compared using the chi-square test or linear-by-linear association. Overall survival (OS) and progression-free survival (PFS) with regard to the ALBI grade and C–P classification were the primary endpoints. Local control (LC) within the RT target volume was also measured. Survival outcomes were compared using a Kaplan–Meier analysis with log-rank tests. Cox proportional hazard models were used to analyze the correlation of survival outcomes with an ALBI grade and C–P classification, along with other relevant possible prognostic factors such as age, sex, performance status, pretreatment tumor markers, and the modified Union for International Cancer Control (mUICC) stage. Statistical significance was set at *p* < 0.05, and all analyses were performed using IBM SPSS Statistics for Windows, Version 24.0 (IBM Corp., Armonk, NY, USA) and GraphPad Prism for Windows, Version 8.4.3 (GraphPad Software, San Diego, CA, USA).

## 3. Results

### 3.1. Patient Characteristics and Liver Function Assessment

A total of 73 patients were included in this study and clinical characteristics were described in Table 1. The patients were predominantly male with a median age of 62 years (range, 27–88) during the treatment period. B-viral hepatitis was the most common etiology (69.9%), while more than half of the patients were mUICC stage IVA, reflecting a large tumor size with vascular invasion in many cases. The common reasons for RT administration after TACE include a vascular invasion of HCC, mostly via the portal vein, incomplete TACE, and regional lymph node metastasis. A detailed summary of the causes and sites of RT after TACE is provided in Table 2.

Thirty-three (45.2%) and forty (54.8%) patients were ALBI grades 1 and 2–3, respectively. The ALBI scores ranged from −3.29 to −0.89 (median −2.44). Patient and disease characteristics for all patients and subsets of patients classified using ALBI grades with the comparison between them are summarized in Table 1. Among the 73 patients, 64 (87.7%) and 9 (12.3%) were classified as C–P classes A and B, respectively. The difference in the distribution of patients according to the ALBI grade and C–P classification was statistically significant (*p* = 0.003). Even though all 33 patients with ALBI grade 1 were classified into C–P class A, the majority of ALBI grade 2–3 patients were also classified into C–P class A. Figure 1 displays the proportion of patients in terms of C–P class for each ALBI grade and vice versa. The detailed relationships of the two classification systems with ALBI and C–P scores are summarized in Appendix A. Other than C–P classes, the Eastern Cooperative Oncology Group (ECOG) performance status and pre-TACE alpha-fetoprotein showed a statistically significant difference between the two subgroups of patients with *p*-values of 0.040 and 0.029, respectively.

### 3.2. Survival Outcomes

Median PFS and OS for all patients were 6.3 months and 23.4 months, respectively, after the median follow-up period of 16.3 months (range, 3.6–107.5). Figure 2 displays the PFS and OS according to the ALBI grades and C–P classes. The median PFS and OS for ALBI grade 1 vs. 2–3 were 8.6 months vs. 5.0 months (*p* = 0.016) and 27.0 months vs. 15.9 months (*p* = 0.006), respectively. Contrastingly, the median PFS and OS for C–P class A vs. B were 6.3 months vs. 6.1 months (*p* = 0.265) and 24.8 months vs. 19.0 months (*p* = 0.630), respectively. Among the multiple potential prognostic factors, ALBI grades 2–3 showed a statistically significant association with worse PFS and OS compared to ALBI grade 1 (*p* = 0.035; hazard ratio [HR] 1.728, 95% confidence interval [CI], 1.038–2.875 for PFS; *p* = 0.021, HR 2.161, 95% CI 1.124–4.156 for OS). Patients with mUICC stage 4A were significantly associated with worse OS compared to patients with mUICC stages 1–3 in the multivariate analysis (*p* = 0.009, HR 2.472, 95% CI 1.251–4.885). No other variables were significantly associated with worse OS in the multivariate analysis. Table 3 summarizes the results of univariate and multivariate Cox regression analyses. The ALBI score as a continuous variable also showed a statistically significant correlation with PFS and OS when analyzed with identical variables in Table 2 (*p* = 0.010, HR 2.073, 95% CI 1.188–3.617 for PFS; *p* = 0.007, HR 2.463, 95% CI 1.278–4.749 for OS), indicating deteriorated survival outcomes with an increasing ALBI score. The results are shown in Appendix A.

The median biologically effective dose using an alpha/beta ratio of 10 (BED10) was 47.3 Gy. Figure 3 shows the difference in the LC based on BED10. Patients receiving BED10 > 47.3 Gy had a median LC of 11.6 months within the RT target volume, whereas patients receiving BED10 ≤ 47.3 Gy had a median LC of 6.6 months (*p* = 0.015).

### 3.3. Toxicity

Three patients developed non-classic RILD, whereas none of the patients experienced classic RILD. RT dose prescriptions were 30 Gy in 10 fractions for one patient and 40 Gy in 10 fractions for two patients. Acute severe exacerbations of liver function during and after treatment, in terms of the ALBI grade, were not significant. Among the 33 patients with ALBI grade 1 before treatment, 21 patients still showed ALBI grade 1 at 1–2 months after the treatment, and 35 of 40 patients with ALBI grade 2–3 showed no worsening in ALBI grade. Appendix A summarizes the detailed pattern of change in ALBI grade after treatment.

## 4. Discussion

This study suggests the role of the ALBI grade as a potential prognosticator in patients administered TACE combined with RT. The ALBI grade has been proven to reflect liver function not only in chronic liver disease but also in patients with HCC, with clearer survival discrimination than that of the C–P class [11]. Conventionally, one of the most widely used liver function indicators, the C–P classification, is usually a critical factor in treatment decision making in patients with HCC [25]. However, the C–P class includes clinical, non-standardized parameters (ascites and encephalopathy) and mutually related factors (serum albumin and ascites) and has no statistical foundation [26]. Given that the ALBI scoring system uses only objective parameters (serum albumin and bilirubin) and is an attractive alternative to the C–P class, it has been evaluated in multiple studies. A large, multicenter retrospective study with 2426 patients with HCC validated the ALBI grade as a significant prognostic index after surgical resection, TACE, and sorafenib across all BCLC stages [12]. Several other studies have also reported the efficacy of the ALBI grade in survival prediction after various treatment modalities, such as surgery, radiofrequency ablation, TACE, stereotactic body radiotherapy (SBRT), and even immunotherapy agents. Moreover, many of these studies showed the superiority of the ALBI grade over C–P class [27]. In the current study, the median PFS and OS for ALBI grade 1 patients were 8.6 months and 27.0 months, respectively, and the median PFS and OS for C–P class A patients were 6.3 months and 24.8 months, respectively. The median PFS and OS for ALBI grade 2–3 patients were 5.0 months and 15.9 months, respectively, and the median PFS and OS for C–P class B patients were 6.1 months and 19.0 months, respectively. These results suggest that ALBI grade 1 patients might have better PFS and OS than C–P class A patients, and ALBI grade 2–3 patients might have worse PFS and OS than C–P class B patients, although it requires validation in the larger population.

Besides representing liver function, the ALBI grade may also reflect the immune-related status of the patient, which may be attributed to tumor progression. For instance, hypoalbuminemia is related to increased vascular permeability and interstitial volume, which are hallmarks of inflammation that can induce cancer growth [28]. In addition, bilirubin showed a suppressive effect on CD4+ T cell reactivity, with significant inhibition of antigen-specific and polyclonal T cell responses [29]. Therefore, an inferior ALBI grade, which is a combination of hypoalbuminemia and hyperbilirubinemia, may suggest not only impaired liver function but also the poor immunologic status of the patient. Recent studies have shown an association between a worse pretreatment ALBI grade and poorer survival outcomes, including both OS and PFS, in patients with non-small cell lung cancer treated with immune checkpoint inhibitors [30,31]. Even though the possible association between the immunologic aspect of the ALBI grade and tumor progression is beyond the scope of the current study, further studies regarding this might be beneficial for the management of various cancers, including HCC.

Although there is an abundance of ALBI-related literature in many treatment modalities, the evaluation of the ALBI grade in patients who underwent TACE and RT collectively was scarce. Advances in RT techniques reduce toxicity by sparing the normal liver while delivering appropriate doses to the tumor and enable wider utilization of RT in patients with HCC [32,33]. Given that RT is usually reserved for unresectable HCC, while TACE is considered one of the first-line treatment modalities for unresectable HCC, their combination has been used in many clinical settings [34,35,36]. In addition, a recent meta-analysis showed the possible therapeutic benefit of TACE plus RT over TACE alone in patients with unresectable HCC [37], suggesting the possible expansion of its usage in suitable patients. In our study, most patients required TACE and RT collectively, owing to the tumor thrombus or incomplete TACE. These patients who are deemed to have unresectable advanced HCC could have deteriorated liver function; therefore, proper evaluation of liver function must precede treatment decision making. In this regard, the significance of the ALBI grade in the setting of TACE plus RT, as well as other modalities, deserves attention. Moreover, owing to its potential association with treatment-related toxicity after both TACE and RT, the pretreatment ALBI grade needs further evaluation in terms of the risk of complications. In a retrospective study of 123 patients, pretreatment ALBI grade III was an independent acute-on-chronic liver failure predictor and was associated with severe complications after TACE [38]. In addition, Lo et al. reported that 18.3% of patients with an ALBI score of ≥−2.76 developed RILD, while only 2.4% of patients with an ALBI score of <−2.76 developed RILD post SBRT for HCC [39]. Therefore, a cautious approach with proper liver function assessment prior to RT dose escalation is necessary, given that a higher RT dose is a double-edged sword with a better chance of LC with an increased risk of toxicity.

This study has some limitations in conjunction with the abovementioned arguments. Owing to its retrospective nature, possible prognostic factors, including the reason for combining RT with TACE (vascular invasion, incomplete TACE, large tumor size, etc.), the interval between TACE and RT, and RT dose and modality, were not controlled. While these factors had no significant effect on survival outcomes in the current study (data not shown), possibly because of the relatively small number of patients without much variation, further research is necessary. However, we were able to show that other possible factors, such as etiology, underlying liver cirrhosis, and previous hepatectomy history were not statistically significantly different between ALBI grade 1 and ALBI grade 2–3. Also, the ALBI grade was the only factor that showed a statistically significant association with both PFS and OS, while the C–P class did not in the multivariate Cox analysis and Kaplan–Meier analysis. Another limitation is the small number of ALBI grade 3 patients, which leads to the grouping of ALBI grade 2 and 3 together. Only one ALBI grade 3 patient was included in the current study, making it difficult to evaluate the significance of severe liver function deterioration in terms of the ALBI grade. Still, given the close relationship between the treatment decision making and the liver function of the patient, it is not likely to use combined TACE and RT in ALBI grade 3 patients. This could suggest that the number of ALBI grade 3 patients does not severely impair the conclusion of the current study. ALBI grade 3 is a relative contraindication of RT in our institutional policy, and RT is applied only in cases that require prompt palliation of severe portal hypertension. In addition, although there was no statistically significant difference in the RT dose between patients with ALBI grade 1 vs. ALBI grade 2–3, a slightly higher BED10 was prescribed in those with ALBI grade 1. The effect of RT dose escalation in combination with TACE may require further prospective studies. 

In conclusion, this study showed that the ALBI grade could better discriminate liver function and predict survival outcomes than that of the C–P classification in patients with HCC treated with TACE plus RT. Reckoning many clinical situations, such as a vessel-invading HCC or incomplete TACE due to the large tumor size or poor vasculature that require TACE and RT collectively, further validation of the ALBI grade as a prognosticator under these circumstances is warranted.

## Figures and Tables

**Figure 1 jpm-13-00354-f001:**
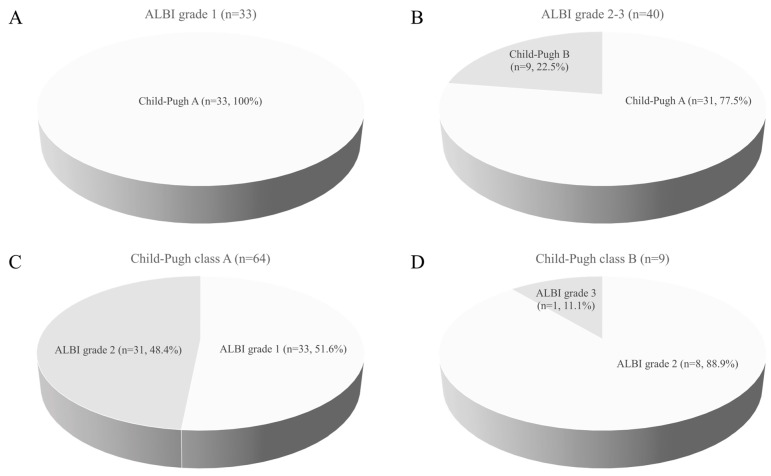
Pie charts show proportions of patients with regard to C–P class in (**A**) ALBI grade 1 patients and (**B**) ALBI grade 2–3 patients. Conversely, the portion of patients according to ALBI grades are shown within (**C**) C–P class A and (**D**) C–P class B. Even though ALBI grade 1 and C–P class B solely consisted of C–P class A and ALBI grade 2–3, respectively, the overlap of ALBI grade 2–3 and C–P class A was identified.

**Figure 2 jpm-13-00354-f002:**
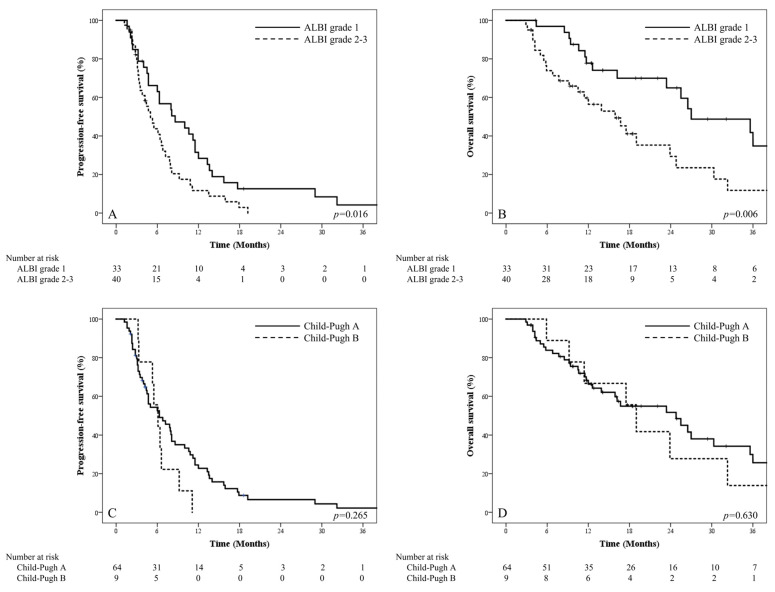
Kaplan–Meier survival curves of PFS and OS according to ALBI grade and C–P class with the number at risk are generated. The two-year PFS rate and OS rate for ALBI grade 1 and ALBI grade 2–3 were (**A**) 12.8% and 0.0%, respectively, and (**B**) 65.0% and 29.4%, respectively. The two-year PFS rate and OS rate for C–P class A and C–P class B were (**C**) 6.6% and 0.0%, respectively, and (**D**) 51.6% and 27.8%, respectively.

**Figure 3 jpm-13-00354-f003:**
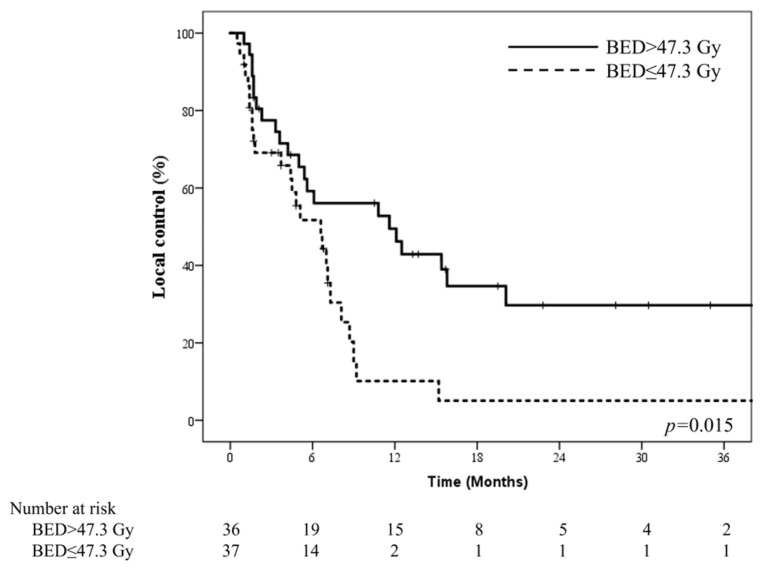
The difference in LC regarding BED10 of 47.3 Gy (equivalent to 35 Gy in 10 fractions) suggests the plausibility of dose escalation in patients who require RT with well-preserved liver function.

**Table 1 jpm-13-00354-t001:** Patient and tumor characteristics.

	All Patients (n = 73)	ALBI Grade 1 (n = 33)	ALBI Grade 2–3 (n = 40)	ALBI Grade 1 vs. Grade 2–3
Characteristic	No. of Patients (%)/ Median (Range)	No. of Patients (%)/ Median (Range)	No. of Patients (%)/ Median (Range)	*p* Value
Gender				1.000
Male	70 (95.9)	32 (97.0)	38 (95.0)	
Female	3 (4.1)	1 (3.0)	2 (5.0)	
Age at TACE+RT (year)	62 (27–88)	64 (26–88)	61 (39–83)	0.807
ECOG				0.040
0–1	56 (76.7)	29 (87.9)	27 (67.5)	
2–3	17 (23.3)	4 (12.1)	13 (32.5)	
Etiology				0.697
B-viral	51 (69.9)	25 (75.8)	26 (65.0)	
C-viral	6 (8.2)	1 (3.0)	5 (12.5)	
Non-B, Non-C	13 (17.8)	5 (15.2)	8 (20.0)	
Alcoholic	3 (4.1)	2 (6.1)	1 (2.5)	
Underlying liver cirrhosis				0.102
Yes	48 (65.8)	25 (75.8)	23 (57.5)	
No	25 (34.2)	8 (24.2)	17 (42.5)	
mUICC stage				0.205
1	1 (1.4)	0 (0.0)	1 (2.5)	
2	7 (9.6)	5 (15.2)	2 (5.0)	
3	23 (31.5)	12 (36.4)	11 (27.5)	
4A	42 (57.5)	16 (48.5)	26 (65.0)	
Pre-TACE AFP (ng/mL)	80.3 (1.3–60,500.0)	14.1 (1.3–24,707.0)	129.6 (2.1–60,500.0)	0.029
Pre-TACE PIVKA-II (mAU/mL)	496.5 (13.0–300,000.0)	397.0 (13.0–75,000.0)	554.0 (15.0–300,000.0)	0.570
Previous resection				0.233
Yes	7 (9.6)	5 (15.2)	2 (5.0)	
No	66 (90.4)	28 (84.8)	38 (95.0)	
No. of previous TACE &/or RFA				0.473
0	33 (45.2)	12 (36.4)	21 (52.5)	
1	17 (23.3)	8 (27.3)	9 (22.5)	
2	6 (8.2)	3 (9.1)	3 (7.5)	
3	4 (5.5)	4 (12.1)	0 (0.0)	
≥4	13 (17.8)	6 (18.2)	7 (17.5)	
Pre-TACE Child-Pugh grade				0.003
A	64 (87.7)	33 (100.0)	31 (77.5)	
B	9 (12.3)	0 (0.0)	9 (22.5)	
RT dose (Total) (Gy)	35.0 (15.0–54.0)	40.0 (20.0–48.0)	35.0 (15.0–54.0)	0.416
RT dose (BED_10_) (Gy)	47.3 (19.5–105.6)	56.0 (30.0–105.6)	47.3 (19.5–70.2)	0.084
TACE-RT interval (weeks)	6.6 (1.0–21.7)	7.4 (2.0–20.0)	6.1 (1.0–21.7)	0.395

Abbreviations: TACE, Transarterial chemoembolization; RT, radiotherapy; ECOG, Eastern Cooperative Oncology Group; mUICC, modified Union for International Cancer Control; AFP, alpha-fetoprotein; PIVKA-II, prothrombin induced by vitamin K absence-II; RFA, radiofrequency ablation; BED_10_, biologically effective dose using an alpha/beta ratio of 10.

**Table 2 jpm-13-00354-t002:** Causes of RT administration and specifications of RT sites.

Causes	No. of patients (%)
PVTT	29 (39.7)
Incomplete TACE	24 (32.9)
Regional LN metastasis	4 (5.5)
Other vessel invasions	3 (4.1)
Combination of the above-mentioned causes	13 (17.8)
RT sites	No. of patients (%)
Tumor thrombus	31 (42.5)
Liver parenchyma	28 (38.4)
Regional LN alone	1 (1.4)
Liver parenchyma + Tumor thrombi	9 (12.3)
Liver parenchyma + Regional LNs	3 (4.1)
Liver parenchyma + Regional LN + Tumor thrombus	1 (1.4)

Abbreviations: TACE, Transarterial chemoembolization; RT, radiotherapy; PVTT, portal vein tumor thrombus; LN, lymph node.

**Table 3 jpm-13-00354-t003:** Univariate and multivariate Cox proportional hazards associations between clinical factors and survival outcomes.

	Progression-Free Survival				Overall Survival			
	UVA		MVA		UVA		MVA	
Variable	Hazard Ratio (95% CI)	*p* Value	Hazard Ratio (95% CI)	*p* Value	Hazard Ratio (95% CI)	*p* Value	Hazard Ratio (95% CI)	*p* Value
Age at TACE+RT ≤60 vs. >60	0.571 (0.338–0.964)	0.036	0.670 (0.387–1.162)	0.154	1.029 (0.549–1.927)	0.930	-	-
Gender	1.122 (0.346–3.632)	0.848	-	-	2.077 (0.633–6.818)	0.228	-	-
ECOG 0–1 vs. 2–3	1.201 (0.680–2.122)	0.527	-	-	1.399 (0.696–2.818)	0.346	-	-
mUICC stage 1–3 vs. 4A	1.640 (0.997–2.698)	0.051	1.394 (0.825–2.354)	0.215	2.703 (1.374–5.319)	0.004	2.472 (1.251–4.885)	0.009
Pre-TACE AFP (ng/mL) ≤80 vs. >80	1.518 (0.921–2.503)	0.102	-	-	1.108 (0.597–2.059)	0.745	-	-
Pre-TACE PIVKA-II (mAU/mL) ≤496.5 vs. >496.5	1.305 (0.748–2.278)	0.349	-	-	1.150 (0.558–2.370)	0.705	-	-
Pre-TACE Child-Pugh grade	1.503 (0.727–3.107)	0.272	-	-	1.222 (0.540–2.768)	0.631	-	-
Pre-TACE ALBI grade 1 vs. 2–3	1.840 (1.111–3.049)	0.018	1.728 (1.038–2.875)	0.035	2.411 (1.256–4.628)	0.008	2.161 (1.124–4.156)	0.021

Abbreviations: UMA, univariate analyses; MVA, multivariate analyses; CI, confidence interval; TACE, transarterial chemoembolization; RT, radiotherapy; ECOG, Eastern Cooperative Oncology Group; mUICC, modified Union for International Cancer Control; AFP, alpha-fetoprotein; PIVKA-II, prothrombin induced by vitamin K absence-II; ALBI, albumin–bilirubin.

## Data Availability

For restrictions on sharing data publicly, data cannot be shared publicly because of potentially identifying or sensitive patient information. Data are available from our Institutional Review Board for researchers who meet the criteria for access to confidential data.

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
