# Peer review of "Evaluation of Pretreatment Albumin–Bilirubin Grade as a Better Prognostic Factor Compared to Child–Pugh Classification in Patients with Hepatocellular Carcinoma Receiving Transarterial Chemoembolization Combined with Radiotherapy"

_jpm, 2023, doi:10.3390/jpm13020354_

Round 1

Reviewer 1 Report

The article's authors aimed to compare the survival outcomes of the patients with HCC who received TACE and TR retrospectively between January 2011 and December 2020 by using ALBI grade and Child-Pugh (C-P) classification. Statistical methods were used in the article, but no evaluation was made by looking at p-values. Therefore, the reason these p-values are high has to be explained in Tab. 1, 2, and Fig. 2 should also be explained.

Author Response

Response 1: Thank you for the comment. In Table 1, as we focused on the value of the ALBI grade in anticipating survival outcomes, we tried to show that other possible factors were not significantly different between each group (ALBI grade 1 vs 2-3). Even though ECOG grade and C-P classification showed a statistically significant difference, which seems quite natural given the function of ALBI grade as a liver function assessment, other possible factors showed no statistically significant difference. It is also similar in Table 2, showing that ALBI grade only had a statistical significance in both progression-free survival and overall survival while other factors did not. The high p-value in Figure 2(C) and (D) also meant that C-P classification did not differentiate survival outcomes as well as ALBI grade. Therefore, we would like to revise our manuscript as follows. Please note that the other part of the manuscript has been changed as well according to the comment from another reviewer.

Original: Discussion

This study has some limitations in conjunction with the abovementioned arguments. Owing to its retrospective nature, possible prognostic factors, including the reason for combining RT with TACE (vascular invasion, incomplete TACE, large tumor size, etc.), the interval between TACE and RT, and RT dose and modality were not controlled. While these factors had no significant effect on survival outcomes in the current study (data not shown), possibly because of the relatively small number of patients without much variation, further research is necessary. In addition, although there was no sta-tistically significant difference in RT dose between patients with ALBI grade 1 vs. ALBI grade 2–3, a slightly higher BED10 was prescribed in those with ALBI grade 1. The effect of RT dose escalation in combination with TACE may require further prospective studies.

Revised:

This study has some limitations in conjunction with the abovementioned arguments. Owing to its retrospective nature, possible prognostic factors, including the reason for combining RT with TACE (vascular invasion, incomplete TACE, large tumor size, etc.), the interval between TACE and RT, and RT dose and modality were not controlled. While these factors had no significant effect on survival outcomes in the current study (data not shown), possibly because of the relatively small number of patients without much variation, further research is necessary. However, we were able to show that other possible factors, such as etiology, underlying liver cirrhosis, and previous hepatectomy history were not statistically significantly different between ALBI grade 1 and ALBI grade 2-3. Also, ALBI grade was the only factor that showed a statistically significant association with both PFS and OS while C-P class did not in multivariate Cox analysis and Kaplan–Meier analysis. Another limitation is the small number of ALBI grade 3 patients which leads to the grouping of ALBI grade 2 and 3 together. Only one ALBI grade 3 patient was included in the current study, making it difficult to evaluate the significance of severe liver function deterioration in terms of ALBI grade. Still, given the close relationship between the treatment decision-making and the liver function of the patient, it is not likely to use combined TACE and RT in ALBI grade 3 patients. This could suggest that the number of ALBI grade 3 patients does not severely impair the conclusion of the current study. ALBI grade 3 is a relative contraindication in our institutional policy, and RT is applied only in cases that require prompt palliation of severe portal hypertension. In addition, although there was no statistically significant difference in RT dose between patients with ALBI grade 1 vs. ALBI grade 2–3, a slightly higher BED10 was prescribed in those with ALBI grade 1. The effect of RT dose escalation in combination with TACE may require further prospective studies.

Reviewer 2 Report

Journal:JPM (ISSN 2075-4426)

Manuscript ID:jpm-2189206

Type:Article

Title:Evaluation of Pretreatment Albumin-Bilirubin Grade as a Better Prognostic Factor Compared to Child-Pugh Classification in Patients with Hepatocellular Carcinoma Receiving Transarterial Chemoembolization Combined with Radiotherapy

The author of this manuscript evaluated the use of pretreatment albumin-bilirubin (ALBI) grade as a prognostic factor in patients with hepatocellular carcinoma (HCC) receiving combined transarterial chemoembolization (TACE) and radiotherapy (RT) and conclused that the ALBI grade could be a prognosticator in HCC patients who treated with combined TACE-RT.

The subject of this manuscript is of value, but there are defects need to be modified.

1. The author should discuss and analyze the statistical methods and results of comparing the two evaluation systems (ALBI grade and Child-Pugh classification) based on the actual progression-free survival (PFS) and overall survival (OS) , so as to clearly show the advantages of the AILB grade.

2. Method section: The scheme of combining TACE and RT should be described in more detail. If TACE is performed first and then RT is performed, the clinical situations require TACE and RT collectively should be described and discussed. 

3. Results section:Thirty-three (45.2%) and 40 (54.8 %) patients were ALBI grades 1 and 2–3, respectively. The author divided the patients into two groups:ALBI grades 1 and 2–3. But how many cases of ALBI grade 2 and ALBI grade 3 in group B, respectively. The reason for this grouping method and its impact on the results should be discussed. 

4. Please check the symbol between grades and with in Line 130: Patient and disease characteristics for all patients and subsets of patients classified by ALBI grades ¬¬with com......

Author Response

Response 1: Thank you for your valuable comment. While the median PFS and OS for ALBI grade 1 patients were 8.6 months and 27.0 months, respectively, the median PFS and OS for C-P class A patients were 6.3 months and 24.8 months, respectively. The median PFS and OS for ALBI grade 2-3 patients were 5.0 months and 15.9 months, respectively, and the median PFS and OS for C-P class B patients were 6.1 months and 19.0 months, respectively. These results somehow suggest that ALBI grade 1 patients had better PFS and OS than C-P class A patients, and ALBI grade 2-3 patients had worse PFS and OS than C-P class B patients. However, it is not available to use Kaplan-Meier methods to compare ALBI grade and C-P class directly as a significant portion of patients overlap between each classification system. Therefore, we used a Cox analysis to assess the association between survival outcomes and each grading system, which showed a significant association between ALBI grade and both PFS and OS while no association was shown with C-P class. We would like to revise our manuscript by adding a comparison between actual PFS and OS between each classification as follows.

Original: Line 213

Moreover, many of these studies showed the superiority of ALBI grade over C-P class [27].

Revised:

Moreover, many of these studies showed the superiority of ALBI grade over C-P class [27]. In the current study, the median PFS and OS for ALBI grade 1 patients were 8.6 months and 27.0 months, respectively, and the median PFS and OS for C-P class A patients were 6.3 months and 24.8 months, respectively. The median PFS and OS for ALBI grade 2-3 patients were 5.0 months and 15.9 months, respectively, and the median PFS and OS for C-P class B patients were 6.1 months and 19.0 months, respectively. These results suggest that ALBI grade 1 patients might have better PFS and OS than C-P class A patients, and ALBI grade 2-3 patients might have worse PFS and OS than C-P class B patients, although it requires validation in the larger population. 

Response 2: Thank you for the comment. According to your suggestion, we would like to move Supplementary Table S1 into the main manuscript so that readers could have a better understandings in clinical scenarios which require TACE and RT collectively. We would like to revise our manuscript as follows.

Original: Line 125

The common reasons for RT administration after TACE include vascular invasion of HCC, mostly the portal vein, incomplete TACE, and regional lymph node metastasis. A detailed summary of the causes and sites of RT after TACE is provided in Supplementary Table S1.

Revised:

The common reasons for RT administration after TACE include vascular invasion of HCC, mostly the portal vein, incomplete TACE, and regional lymph node metastasis. A detailed summary of the causes and sites of RT after TACE is provided in Table 2.

Original: Line 233

Given that RT is usually reserved for unresectable HCC, while TACE is considered one of the first-line treatment modalities for unresectable HCC, their combination has been used in many clinical settings [34-36]. In addition, a recent meta-analysis showed the possible therapeutic benefit of TACE plus RT over TACE alone in patients with unresectable HCC [37], suggesting the possible expansion of its usage in suitable patients.

Revised:

Given that RT is usually reserved for unresectable HCC, while TACE is considered one of the first-line treatment modalities for unresectable HCC, their combination has been used in many clinical settings [34-36]. In addition, a recent meta-analysis showed the possible therapeutic benefit of TACE plus RT over TACE alone in patients with unresectable HCC [37], suggesting the possible expansion of its usage in suitable patients. In our study, most patients required TACE and RT collectively owing to the tumor thrombus or incomplete TACE. These patients who are deemed to have unresectable advanced HCC could have deteriorated liver function; therefore, proper evaluation of liver function must precede treatment decision-making.

Response 3: Thank you for the comment. This grouping method was somewhat inevitable as only one ALBI grade 3 patient was included in the current study, as we showed in Figure 1(D) and Supplementary Table S3. Given that patients with severely deteriorated liver function would not be a candidate for aggressive treatments like TACE and RT, there was no C-P class C patient in the current study. If the ALBI grade 3 patient had been classified as C-P class C when the treatment decision was made, we think this patient would have not been treated by TACE and RT. We would like to revise our manuscript as follows. Please note that the other part of the manuscript has been changed as well according to the comment from another reviewer.

Original: Line 254

While these factors had no significant effect on survival outcomes in the current study (data not shown), possibly because of the relatively small number of patients without much variation, further research is necessary. However, we were able to show that other possible factors, such as etiology, underlying liver cirrhosis, and previous hepatectomy history were not statistically significantly different between ALBI grade 1 and ALBI grade 2-3. Also, ALBI grade was the only factor that showed a statistically significant association with both PFS and OS while C-P class did not in multivariate Cox analysis and Kaplan–Meier analysis. Another limitation is the small number of ALBI grade 3 patients which leads to the grouping of ALBI grade 2 and 3 together. Only one ALBI grade 3 patient was included in the current study, making it difficult to evaluate the significance of severe liver function deterioration in terms of ALBI grade. Still, given the close relationship between the treatment decision-making and the liver function of the patient, it is not likely to use combined TACE and RT in ALBI grade 3 patients. This could suggest that the number of ALBI grade 3 patients does not severely impair the conclusion of the current study. ALBI grade 3 is a relative contraindication in our institutional policy, and RT is applied only in cases that require prompt palliation of severe portal hypertension.

Revised:

While these factors had no significant effect on survival outcomes in the current study (data not shown), possibly because of the relatively small number of patients without much variation, further research is necessary.

Response 4: Thank you for the comment. We correct the error by removing the typo as follows.
